# Low Intensity Extracorporeal Shock Wave Therapy as a Novel Treatment for Stress Urinary Incontinence: A Randomized-Controlled Clinical Study

**DOI:** 10.3390/medicina57090947

**Published:** 2021-09-08

**Authors:** Kun-Ling Lin, Kuang-Shun Chueh, Jian-He Lu, Shu-Mien Chuang, Bin-Nan Wu, Yung-Chin Lee, Yi-Hsuan Wu, Mei-Chen Shen, Ting-Wei Sun, Cheng-Yu Long, Yung-Shun Juan

**Affiliations:** 1Graduate Institute of Clinical Medicine, College of Medicine, Kaohsiung Medical University, Kaohsiung 807, Taiwan; nancylin95@gmail.com (K.-L.L.); spacejason69@yahoo.com.tw (K.-S.C.); maivy0314@gmail.com (Y.-H.W.); 2Department of Obstetrics and Gynecology, Kaohsiung Medical University Hospital, Kaohsiung 807, Taiwan; 3Department of Obstetrics and Gynecology, College of Medicine, Kaohsiung Medical University, Kaohsiung 807, Taiwan; 4Department of Urology, Kaohsiung Medical University Hospital, Kaohsiung 807, Taiwan; u9181002@gmail.com (S.-M.C.); leeyc12345@yahoo.com.tw (Y.-C.L.); bear5824@gmail.com (M.-C.S.); selina750220@yahoo.com.tw (T.-W.S.); 5Department of Urology, Kaohsiung Municipal Ta-Tung Hospital, Kaohsiung 807, Taiwan; 6Emerging Compounds Research Center, Department of Environmental Science and Engineering, College of Engineering, National Pingtung University of Science and Technology, Pingtung County 912, Taiwan; toddherpuma@yahoo.com.tw; 7Department of Urology, College of Medicine, Kaohsiung Medical University, Kaohsiung 807, Taiwan; 8Department of Pharmacology, Graduate Institute of Medicine, College of Medicine, Kaohsiung Medical University, Kaohsiung 807, Taiwan; binnan@kmu.edu.tw; 9Department of Urology, Kaohsiung Municipal Hsiao-Kang Hospital, Kaohsiung 807, Taiwan; 10Regenerative Medicine and Cell Therapy Research Center, Kaohsiung Medical University, Kaohsiung 807, Taiwan

**Keywords:** low intensity extracorporeal shock wave, stress urinary incontinence, urine leakage, overactive bladder, validated standardized questionnaires

## Abstract

*Background and Objectives*: To evaluate the effects of low intensity extracorporeal shock wave therapy (LiESWT) on stress urinary incontinence (SUI). *Materials and Methods*: This investigation was a multicenter, single-blind, randomized-controlled trial study. Sixty female SUI patients were randomly assigned to receive LiESWT with 0.25 mJ/mm^2^ intensity, 3000 pulses, and 3 pulses/s, once weekly for a 4-week (W4) and 8-week (W8) period, or an identical sham LiESWT treatment without energy transmission. The primary endpoint was the changes in urine leakage as measured by a pad test and validated standardized questionnaires, while the secondary endpoint was the changes in a 3-day urinary diary among the baseline (W0), the W4 and W8 of LiESWT, and 1-month (F1), 3-month (F3), and 6-month (F6) follow-up after LiESWT. *Results*: The results showed that 4 weeks of LiESWT could significantly decrease urine leakage based on the pad test and validated standardized questionnaire scores, as compared to the sham group. Moreover, 8 weeks of LiESWT could significantly reduce urine leakage but increase urine volume and attenuate urgency symptoms, which showed meaningful and persistent improvement at W8, F1, F3, and F6. Furthermore, validated standardized questionnaire scores were significantly improved at W8, F1, F3, and F6 as compared to the baseline (W0). *Conclusions*: Eight weeks of LiESWT attenuated SUI symptoms upon physical activity, reduced urine leakage, and ameliorated overactive bladder symptoms, which implied that LiESWT significantly improved the quality of life. Our findings suggested that LiESWT could serve as a potentially novel and non-invasive treatment for SUI.

## 1. Introduction

Stress urinary incontinence (SUI), according to the International Continence Society, is the involuntary leakage of urine upon physical exertion, effort, coughing, or sneezing [1]. The global prevalence of urinary incontinence in the female population is approximately 50% [2], which is a significant problem, namely the impairment of quality of life (QoL), such as reducing social interaction and physical activity, destroying sexual relationships, and worsening emotional and mental health.

Excessive hypermobility of the urethra and/or bladder neck resulting from pelvic floor degeneration and internal sphincter deficiency are the two main mechanisms of SUI. Clinical managements of SUI are various, such as lifestyle changes, pelvic floor muscle training (PFMT), and medication, as well as surgery. PFMT is advocated as a valuable first choice with negligible complications and an acceptable cure rate, but it requires patient long-term regular exercise to achieve satisfying results [3]. Mid-urethral sling (MUS) surgery is considered as the preferred surgical method because of the high cure rate [4], but it has a significant rate of complications [5]. Therefore, novel therapies that can restore normal pelvic floor strength and internal sphincter function are urgently needed.

The extracorporeal shock wave is a longitudinal acoustic wave that propagates through human tissues at the speed of an ultrasound wave in water [6]. As an example of these, low intensity extracorporeal shock wave therapy (LiESWT) could induce local inflammation reaction and promote angiogenesis, recruit mesenchymal stem cells and endothelial progenitor cells, stimulate cellular proliferation and regeneration [7], and inhibit oxidative stress, thus improving blood circulation and enhancing tissue repair [8]. In recent basic and clinical studies, the efficacy of LiESWT in treating chronic injuries of soft tissues (e.g., erectile dysfunction (ED) [9,10,11] and chronic prostatitis/chronic pelvic pain syndrome (CP/CPPS) [12,13] has been well established. More importantly, the benefit of LiESWT is that it is a non-invasive, outpatient-based therapy featuring short treatment sessions, which helps patients to be free from anesthesia and the adverse effects of medication or surgery.

A recent study using a vaginal balloon dilation (VBD)-induced SUI rat model showed that SUI rats treated with LiESWT (0.06 mJ/mm^2^, and 300 shocks at 3 pulses/s) following VBD had significantly higher leak-point pressure (LPP) relative to those receiving VBD only. LiESWT improved LPP and ameliorated SUI by promoting urethral sphincter regeneration, angiogenesis, and progenitor cell recruitment [7]. With the same energy and course of LiESWT, another VBD-induced SUI rat study revealed that an increased amount of urethral and vaginal smooth and striated muscle content, and increased thickness of the vaginal wall, were noted after LiESWT treatment [14]. Higher LPP was also found in the LiESWT group [14]. Recently, we published our preliminary results on human subjects showing that 8 weeks of LiESWT attenuated SUI symptoms with short term follow-up [15]. Interestingly, LiESWT also ameliorated overactive bladder (OAB) symptoms during follow-up [15]. In the current study, we further evaluate the clinical application of LiESWT on SUI patients and its persistence efficacy, including attenuating bladder urine leaks, impacting on overactive bladder (OAB) symptoms, and promoting quality of life.

## 2. Materials and Methods

### 2.1. Design

This single-blind, prospective, randomized-controlled trial was performed at a tertiary medical center in Taiwan between December 2018 and January 2020. The investigation was approved by the Kaohsiung Medical University Hospital Institutional Review Board (IRB No. KMUHIRB-F(II)-20180010) and was registered at clinicaltrials.gov (NCT04059133) on 16 August 2019. The study enrolled 60 female participants aged 20–75 years who had been diagnosed with SUI or mixed urinary incontinence (MUI) but SUI predominant patients for more than 3 months in this investigation (sham, *n* = 15; LiESWT, *n* = 45). The major inclusion and exclusion criteria are shown in Table 1. Based on the limitations regarding the feasibility of the project at our center, as well as a type I error (α) of 0.05 and type II error (β) of 0.2, we aimed to enroll a total of 60 patients in this study. Considering that patients would be more likely to join a trial if they had a good chance of getting active treatment, participants were randomly assigned in a 3:1 ratio to the study groups with blinded computer-assisted allocation. All participants were provided with informed consent before entering the study and randomly allocated to the sham group or to the LiESWT group by computer generated random numbers.

### 2.2. Physical Indicators and Biochemical Parameters of Studied Participants

The physical and serum parameters of metabolic syndrome were associated with the symptoms of SUI [16]. In order to investigate the baseline characteristics of SUI population, the physical indicators (age, height, weight, waistline, body mass index, systolic pressure, diastolic pressure, and mean arterial pressure) and biochemical parameters (hemoglobin A1c (glycated hemoglobin), blood sugar, glutamate oxaloacetate transaminase and glutamate pyruvate transaminase for liver function index, blood urea nitrogen and creatinine for renal function index, lipid profile on triglycerides, cholesterol, high-density lipoprotein, and low-density lipoprotein) were analyzed.

### 2.3. Procedure and Medical Information of LiESWT

Participants were informed of treatment modalities, including the required consent to join this study and once weekly LiESWT for 8 weeks and follow-up at 6-month after completing the course of treatment (Figure 1). Our instrumentation was the DUOLITH SD1-TOP focused shock wave system (STORZ MEDICAL AG). The LiESWT was applied with 0.25 mJ/mm^2^ of intensity, 3000 pulses of shock, and 3 pulses/s of frequency on the middle urethra, including middle, left, and right side of the participant’s labia [15]. The sham group used an air pad probe to block energy transmission to the tissue, but the machine still emitted shock wave generation.

### 2.4. Pad Test for the Evaluation of SUI

The pad test was applied as a non-invasive diagnosis for quantifying the severity of urine leakage in SUI participants [17,18]. The purpose of pad test in this study was to evaluate the effect of LiESWT on reducing urinary incontinence. The detailed steps were performed according to our previous study [14]. The percentage (%) of improvement was evaluated at 4-week (W4), 8-week (W8), and 1-month follow-up (F1), 3-month follow-up (F3), and 6-month follow-up (F6) after LiESWT treatment, and the results were normalized with pretreatment baseline data (W0).

### 2.5. Outcomes Measures and Therapeutic Efficacy Assessment for LiESWT

To analyze the effects of LiESWT, the primary endpoints were the changes in pad test and questionnaires (Overactive Bladder Symptom Score (OABSS), International Consultation on Incontinence Questionnaire Short Form (ICIQ-SF), Urinary Distress Inventory Short Form (UDI-6), and Incontinence Impact Questionnaire Short Form (IIQ-7)) [15], and the secondary endpoints were the changes in 3-day urinary diary (bedtime and wake up time, fluid intake, amount of urine drained, and leaks) at the baseline (W0), 4-week (W4), 8-week (W8), and 1-month follow-up (F1), 3-month follow-up (F3), and 6-month follow-up (F6).

### 2.6. Statistical Analysis

Data were analyzed using SAS 9.3 (SAS Institute, Cary, NC, USA). Quantitative data were represented as the mean ± standard error of mean (SEM). Paired *t*-test and one-way analysis of variance were used to perform a repeated measurement analysis for intragroup before/after treatment [19]. The paired *t*-test was performed in the sham group (W0 vs. W4). The post hoc Tukey’s honestly significant difference tests were used to make comparisons between the LiESWT subgroups and to calculate *p*-values for comparison [19]. The student *t*-test was performed for the intergroup comparison (the sham group vs. the LiESWT group). For all statistical analyses, *p* < 0.05 was statistically significant.

## 3. Results

### 3.1. Diagnoses

From 2018 to January 2020, a total of 60 female participants, aged 20–75 years, were enrolled in this clinical trial study (NCT04059133). The designed timetable is shown in Figure 1. The physical and serum parameters of metabolic syndrome may be associated with the symptoms of SUI. Therefore, we analyzed the baseline values and demographic features of patients. As presented in Table 2, all physical indicators and serum parameters were characterized for the normal range at W0. The results also revealed that there were no significant differences in these parameters between the sham group and the LiESWT group.

### 3.2. Primary and Secondary End Points

A pad test is useful for detecting bladder involuntary leakage of urine during physical activity examination. The amount of bladder urine leakage in the sham group was no significantly different before or after the 4-week treatment (*p* = 0.281) (Table 3 and Figure 2A). However, the 4-week LiESWT group noticeably decreased urine leakage as compared to the sham group (*p* = 0.004). Moreover, LiESWT significantly decreased urine leakage from 7.93 ± 1.32 (W0) to 2.43 ± 0.40 g (W4, *p* < 0.001) and sustained this until F6 (Table 3 and Figure 2B). Further analysis found that 64.4% of participants had moderate to better improvement (>50%) after 4 weeks of LiESWT, and the proportion of improvement increased to 68.8%, 77.8%, 82.3%, and 84.5% at W8, F1, F3, and F6, respectively (Figure 2C).

The 3-day urinary diary was analyzed (Table 3). LiESWT significantly improved the mean times of the urgency at W8 and sustained the improvement until F6. The average urine volume was noticeably raised at F3 and F6. According to the 3-day voiding diary, 8 weeks of LiESWT significantly improved OAB symptoms in those SUI participants.

The effects of LiESWT on OAB symptoms, physical activity induced involuntary urine leakage, and QoL were examined. The results of the questionnaire for QoL and social activity are shown in Figure 3 and Table 3. In the sham group, there was no significant difference in OABSS, ICIQ-SF, UDI-6, and IIQ-7 questionnaire scores. However, 4 weeks of LiESWT significantly improved for all questionnaires, ICIQ-SF (*p* = 0.034), UDI-6 (*p* = 0.040), and IIQ-7 (*p* = 0.048) scores as compared to the sham group, except for OABSS (*p* = 0.520). In the LiESWT group, all questionnaires also revealed significant improvement at W8, F1, F3, and F6 (Figure 3 and Table 3). Based on the questionnaires, the results showed that LiESWT treatment improved both bladder urine leakage and OAB symptoms, including urinary frequency, nocturia, urgency, and urgency incontinence between W8 and F6 as compared to W0 (Figure 3).

### 3.3. Safety and Adverse Effects

All participants in this study were well tolerated with LiESWT. No significant adverse reactions related to LiESWT were detected, such as skin ecchymosis, hematuria, or intolerable pain. There was no single patient withdrawn from the study due to any adverse effect.

## 4. Discussion

This investigation was a multicenter, single-blind, randomized controlled trial study. The results showed that 8 weeks of LiESWT decreased urine leakage as shown by the pad test and attenuated urgency symptoms as shown by the 3-day urinary diary. There was significant and persistent improvement for as long as 6 months of follow-up. Moreover, validated standardized questionnaire scores were also significantly improved at W8, F1, F3, and F6 as compared to the baseline. These findings indicate that 8 weeks of LiESWT significantly attenuated SUI symptoms by reducing urine leakage, ameliorating overactive bladder symptoms, and improving QoL, which implies that LiESWT could serve as a novel, non-invasive treatment for SUI (Figure 4).

Clinically, the applications of LiESWT in urological disorders includes ED [9,10,11] and CP/CPPS [12,13], but the safety and the effectiveness of treating female SUI are still under investigation. LiESWT (0.09–0.25 mJ/mm^2^, 1500–6000 pulses, 1–2 times per week, 4–12 sessions) was found to significantly increase penile hemodynamics and induce penile tissue regeneration in ED patients [9,11], and 50% of these patients maintained such a therapeutic effect after 2 years [10]. In addition, LiESWT with 0.25 mJ/mm^2^ intensity, 2500–3000 pulses, and 3–4 pulses/s, once each week for 4 weeks can noticeably relieve perineum pain, improve bladder voiding symptoms, and QoL; its significant therapeutic effects lasted from 4 months to 1 year posttreatment [12,13,20]. Our results showed that 8 weeks of LiESWT with 0.25 mJ/mm^2^ intensity, 3000 pulses, 3 pulses/s and once weekly could significantly decrease urine leakage, attenuate urgency symptom of overactive bladder, and improve QoL, which showed persistent improvement for as long as 6-month of follow-up. In summary, the application of LiESWT is helpful not only for the treatment of CPPS and ED but also SUI. Further application of LiESWT on other urologic diseases, such as OAB is under investigation.

SUI is a common and disturbing disease with a variety of management modalities, including PFMT, biofeedback training, electrostimulation, vaginal laser therapy, and surgery. PFMT is promoted as a valuable first choice physical therapy for SUI women, with a reported cure rate of 56%, and the therapeutic effect can be maintained for up to one year after treatment [3]. However, there were no standard parameters of muscle contraction and relaxation for PFMT, and it required more than six months to achieve significant improvement; the patients were prone to give up PFMT [21]. Moreover, the effect of PFMT on patients with moderate to severe urinary incontinence is also poor. MUS surgery seemed to work well for short term follow-up to one year in SUI patients with an over 80% cure rate [4]. However, while few in number, implanted slings have surgical risks and adverse consequences. Souders et al. analyzed more than 70,000 legal statements on the use of artificial mesh or sling from 2000 to 2014, and found that most (63%) of them were related to the retropubic or transobturator sling procedures for the treatment of SUI [22]. The rise in lawsuits does not reflect the acceptably low complication rates for slings for SUI reported in the literature. Reported complications included bladder injury (0.8–11.4%), vaginal injury (0.8–15%), hemorrhage (0.7–5.5%), urinary tract infections (0.9–33%), urine retention (0.8–11.4%), and mesh erosion (0–10%) [5]. According to FDA recommendations, clinically meaningful improvement in SUI symptoms is defined as improvement in pad weight or the number of incontinence episodes with a reduction from baseline greater than 50%. Our investigation demonstrated that 64.4% of women had moderate to better improvement (>50%) after 4 weeks of LiESWT therapy and sustained this until F6 (Figure 2C). In addition, no side effects were observed in participants during the LiESWT treatment and follow-up.

Interestingly, the results also showed that 8 weeks of LiESWT with 0.25 mJ/mm^2^ intensity, 3000 pulses, 3 pulses/s and once weekly could not only significantly decrease urine leakage but also attenuate the urgency symptom of the overactive bladder and improve QoL, which showed persistent improvement for as long as 6-month follow-up. The actual mechanism of the beneficial effect of LiESWT on OAB is still unclear. The main reason why LiESWT for SUI can also ameliorate OAB symptoms is that LiESWT can restore pelvic floor function and, thereafter, improve conscious control of the bladder by activating the part of the brain responsible for the voluntary urinary inhibition reflex. In a previous study, the contraction of the urethral sphincter led to the suppression of detrusor pressure. By this mechanism, the voluntary urinary inhibition reflex can be mediated [23,24]. Several studies have also raised the “integral theory”, which points out that the instability and stretch of the pelvic floor muscle induces inadequate micturition reflex and results in OAB symptoms [25]. In our previous Er:YAG vaginal laser treatment for SUI study, OABSS also significantly decreased following vaginal laser treatment [26]. Moreover, another study has also proved that pelvic floor reconstruction could ameliorate OAB symptoms [27]. From these points of view, LiESWT consolidates the pelvic floor muscle, reduces the micturition reflex and, finally, has benefit for OAB symptoms.

The status of pelvic floor muscles plays an important role in pelvic organ study. The main cause of SUI is related to pelvic floor malfunction, which results in pelvic organ over mobilization. Previous studies of animal models have suggested that LiESWT could induce a local inflammation reaction and promote angiogenesis by enhancing the expression of VEGF and the recruitment of mesenchymal stem cells and endothelial progenitor cells to the injured site [7]. LiESWT was also shown to stimulate cellular proliferation and regeneration [7], inhibit oxidative stress production [8], and anti-apoptotic cells [8], thus improving blood circulation [7,8], increasing urethral muscle regeneration [7], and enhancing tissue repair [8]. Theoretically, through the effect of LiESWT, pelvic floor muscles can repair gradually, and the over mobilization of the bladder can attenuate, which results in the improvement of stress urinary incontinence.

Our study had several limitations. First, patients were limited to small numbers. People benefiting from LiESWT treatment have yet to be confirmed in larger prospective studies. Second, the inability to obtain bladder tissue in clinical trials, leading to biomarker analysis (such as inflammation, angiogenesis, tissue repair, and regeneration-related genes), was not comprehensive in this study. Animal experiments are needed in the future to investigate the potential molecular mechanism underlying LiESWT treatment for SUI. Another important limitation of this study was the lack of follow-up for the sham-treated participants who dropped out and refused to follow up for reasons of ineffectiveness. Although it was explained at the time of recruitment that the LiESWT treatment might not be effective immediately, some participants in the sham group, due to over expectations of the efficacy of LiESWT, dropped out and refused to follow up for reasons of ineffectiveness after 5–6 treatments with LiESWT, which resulted in the lack of W8, F1, F3, and F6 data.

## 5. Conclusions

The present study demonstrated that 8 weeks of LiESWT could improve SUI symptoms during physical activity, reduce urine leakage, lessen OAB symptoms, and promote QoL. The effects of LiESWT may restore the pathophysiology of SUI and sustain it for 6 months. These findings suggest that LiESWT could serve as a potentially promising and alternative method for treating SUI patients. Authors should discuss the results and how they can be interpreted from the perspective of previous studies and the working hypotheses. The findings and their implications should be discussed in the broadest context possible. Future research directions may also be highlighted.

## Figures and Tables

**Figure 1 medicina-57-00947-f001:**
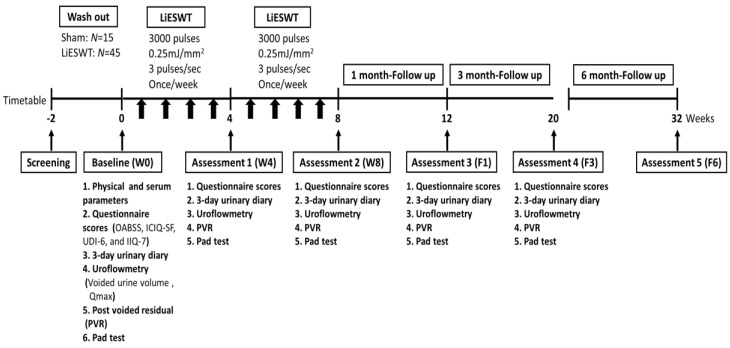
Timetable designed for the clinical trial of stress urinary incontinence (SUI). LiESWT: Low intensity extracorporeal shock wave therapy; W0: baseline; W4: 4-week; W8: 8-week; F1: 1-month; F3: 3-month; F6: 6-month; OABSS: Overactive Bladder Symptom Score; ICIQ-SF: International Consultation on Incontinence Questionnaire Short Form; UDI-6: Urinary Distress Inventory Short Form; IIQ-7: Incontinence Impact Questionnaire Short Form.

**Figure 2 medicina-57-00947-f002:**
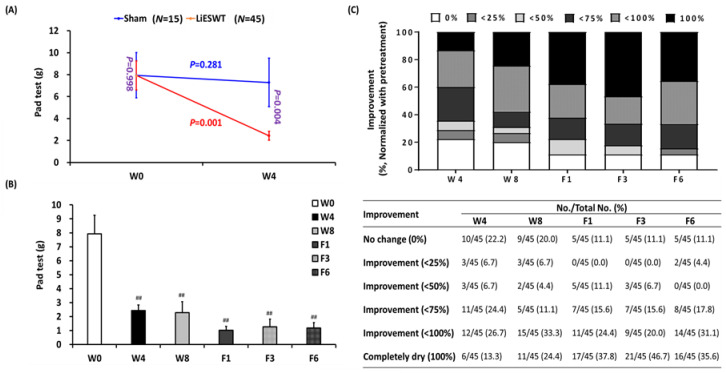
LiESWT decreased bladder leaks by pad test. (**A**) Pad test of the studied population for stress urinary incontinence (SUI) at 4 weeks (W4) in the sham-treated group (*N* = 15) and the LiESWT-treated group (*N* = 45). The blue font or red font denotes the *p*-value before and after 4 weeks of treatment in the sham group or in the LiESWT-treated group. The purple font indicates the *p*-value between the sham-treated group and the LiESWT-treated group at baseline (W0) and W4. Values are reported as means ± SE. (**B**) Pad test of the studied population for SUI at W4, 8 weeks (W8), 1-month follow-up (F1), 3-month follow-up (F3), and 6-month follow-up (F6). Values are reported as means ± SE. *N* = 45. ^##^ *p* < 0.01 compared to the W0. (**C**) The percentage of improvement at W4, W8, F1, F3, and F6 after the LiESWT treatment, as normalized with the W0. *N* = 45.

**Figure 3 medicina-57-00947-f003:**
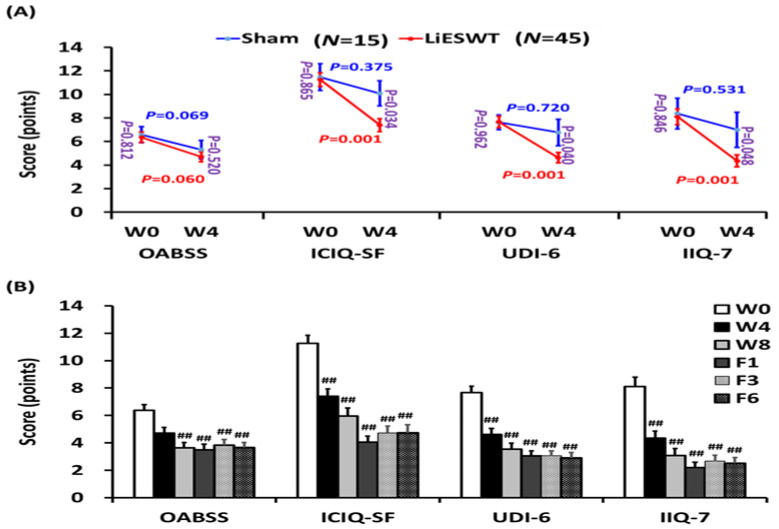
Changes in stress urinary incontinence (SUI) symptoms and validated standardized questionnaire scores after the LiESWT treatment. (**A**) The validated standardized questionnaire scores included the Overactive Bladder Symptom Score (OABSS), International Consultation on Incontinence Questionnaire Short Form (ICIQ-SF), Urogenital Distress Inventory (UDI-6) Short Form, and Incontinence Impact Questionnaire-7 (IIQ-7) score at week 4 (W4) in the sham group (*N* = 15) and the LiESWT-treated group (*N* = 45). The blue font or red font denotes the *p*-value before and after 4 weeks treatment in the sham group or in the LiESWT-treated group, respectively. The purple font indicates the *p*-value between the sham group and the LiESWT-treated group at the baseline (W0) and W4. Values are the means ± SE. (**B**) The OABSS, ICIQ-SF, UDI-6, and IIQ-7 questionnaire scores analyzed at W4, 8 weeks (W8), 1-month follow-up (F1), 3-month follow-up (F3), and 6-month follow-up (F6). Values are the means ± SE. *N* = 45. ^##^ *p* < 0.01 compared to the W0.

**Figure 4 medicina-57-00947-f004:**
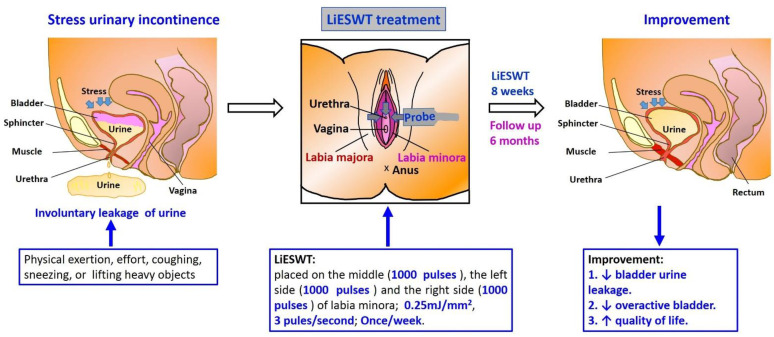
Short graphic abstract of studying the proposed potential effect of LiESWT. SUI, stress urinary incontinence; LiESWT, low intensity extracorporeal low energy shock wave therapy.

**Table 1 medicina-57-00947-t001:** Inclusion and exclusion criteria.

Inclusion Criteria	Exclusion Criteria
Female patients within 18~75 years old. Patients have a ≥3 month history of experiencing Stress Urinary Incontinence (SUI) per week (self reported). Patients can understand and finish the questionnaires. Patients would like to sign the informed consent. Urinary leakage ≥ 2 g.	Urinary tract infection detected at screening and recurrent urinary tract infections (more than 3 episodes in the past 3 months). Comorbidities relevant to OAB (diabetes mellitus, spinal cord injury, stroke, or neurogenic diseases). Severe cardiovascular diseases. Coagulopathy. Liver failure. Renal failure. Chronic urinary inflammation (interstitial cystitis, urethral syndrome, or painful bladder syndrome), drug or alcohol abuse in the past 12 months. Lower urinary tract surgeries in the past 6 months. Perineal operations, intravesical injection, irradiation, shockwave or electrostimulation in the past 12 months. Urinary catheterization, urologic malignancy, gross hematuria, urolithiasis, chronic pelvic pain, or inability to comprehend or comply with instructions.

OAB: overactive bladder.

**Table 2 medicina-57-00947-t002:** Baseline characteristics of stress urinary incontinence (SUI) population.

Parameter	Sham (Mean ± SE)	LiESWT (Mean ± SE)	Range
Physical parameter			
Female age (years)	52.8 ± 1.9	53.8 ± 1.5	20–75
Height (cm)	158.0 ± 1.6	158.9 ± 0.9	
Weight (kg)	59.4 ± 1.9	61.2 ± 1.3	
BMI (kg/m^2^)	23.8 ± 0.7	24.2 ± 0.4	18.5–24
Waistline (cm)	87.2 ± 2.8	85.6 ± 1.4	
Systolic pressure (mmHg)	124.4 ± 3.0	121.4 ± 2.9	100–120
Diastolic pressure (mmHg)	77.9 ± 2.3	74.5 ± 1.7	60–80
MAP	93.4 ± 2.3	90.1 ± 2.0	70–110
Serum parameter			
HbA1C (%)	5.67 ± 0.12	5.63 ± 0.06	4–6
AC sugar (mg/dL)	101.7 ± 3.7	101.0 ± 1.6	65–109
BUN (mg/dl)	12.54 ± 0.80	12.35 ± 0.45	8–20
Creatinine (mg/dL)	0.67 ± 0.03	0.68 ± 0.02	0.44–1.03
GOT(AST) (IU/L)	22.5 ± 1.0	23.5 ± 1.3	10–42
GPT(ALT) (IU/L)	23.8 ± 2.2	23.9 ± 2.2	10–40
Triglycerides (mg/dL)	92.1 ± 12.3	96.9 ± 9.0	35–160
Cholesterol (mg/dL)	214.2 ± 8.2	203.7 ± 6.7	140–200
HDL (mg/dL)	61.5 ± 3.6	57.2 ± 2.2	29–85
LDL (mg/dL)	134.9 ± 8.3	124.9 ± 6.3	0–130

Note: BMI, body mass index; MAP, mean arterial pressure; HbA1C, hemoglobin A1c (glycated hemoglobin); AC, Ante Cibum (before meals); BUN, blood urea nitrogen; GOT, glutamate oxaloacetate transaminase; GPT, glutamate pyruvate transaminase; LDL, low-density lipoprotein; HDL, high-density lipoprotein; Values are means ± SE. *p* < 0.05 vs. sham group. *N* = 15 (Sham) and *N* = 45 (LiESWT).

**Table 3 medicina-57-00947-t003:** Urodynamic parameters and questionnaire score of study population for stress urinary incontinence (SUI).

Parameter	Sham (*N* = 15)		LiESWT (*N* = 45)					
	W0	W4	W0	W4	W8	F1	F3	F6
Pad test (g)	7.94 ± 2.07	7.29 ± 2.21	7.93 ± 1.32	2.43 ± 0.40 ^++,##^	2.29 ± 0.78 ^##^	1.02 ± 0.28 ^##^	1.27 ± 0.54 ^##^	1.18 ± 0.38 ^##^
**3-day urinary diary record**								
Intake (mL)	1698.7 ± 127.7	1703.2 ± 127.8	1757.6 ± 94.3	1822.9 ± 94.8	1738.2 ± 74.7	1678.5 ± 99.4	1680.2 ± 89.8	1785.1 ± 75.9
Output (mL)	1795.3 ± 104.6	1727.4 ± 126.4	1787.6 ± 107.7	1845.1 ± 99.8	1779.9 ± 86.7	1779.5 ± 104.7	1886.2 ± 110.4	1780.4 ± 70.1
Average voided volume (mL)	204.6 ± 15.1	221.4 ± 15.6	212.4 ± 8.9	235.5 ± 9.3	245.4 ± 9.2	245.4 ± 9.9	248.9 ± 8.2 ^#^	249.0 ± 10.2 ^#^
Functional bladder capacity (mL)	376.6 ± 30.9	378.4 ± 30.0	374.3 ± 16.8	393.2 ± 19.7	389.6 ± 18.8	380.3 ± 19.5	400.8 ± 17.6	398.4 ± 19.6
Daytime frequency (times)	8.86 ± 0.50	8.00 ± 0.38	8.54 ± 0.32	8.42 ± 0.33	7.98 ± 0.34	7.91 ± 0.30	8.12 ± 0.29	8.12 ± 0.25
Nocturia (times)	1.00 ± 0.13	0.81 ± 0.17	0.81 ± 0.10	0.82 ± 0.13	0.69 ± 0.12	0.59 ± 0.12	0.61 ± 0.09	0.55 ± 0.11
**Urgency (times)**	1.86 ± 0.46	1.68 ± 0.51	2.09 ± 0.33	1.32 ± 0.24	1.06 ± 0.22 ^#^	0.57 ± 0.15 ^##^	0.86 ± 0.19 ^##^	1.07 ± 0.21 ^#^
**Uroflowmetry data**								
Voided urine volume (mL)	367.2 ± 38.6	378.9 ± 34.9	346.4 ± 24.0	355.5 ± 25.8	394.3 ± 24.4	427.0 ± 23.9 ^#^	433.6 ± 26.4 ^#^	392.0 ± 24.7
Maximum flow rate (mL/s)	32.2 ± 3.3	33.7 ± 2.9	30.1 ± 1.7	31.2 ± 2.1	33.7 ± 2.4	38.0 ± 2.0 ^#^	36.3 ± 2.2	33.1 ± 1.9
PVR (mL)	50.7 ± 11.5	44.3 ± 9.7	53.0 ± 5.6	35.7 ± 4.3	28.3 ± 4.6 ^#^	27.5 ± 4.1 ^#^	26.7 ± 5.0 ^#^	33.5 ± 5.6
**Questionnaire score (points)**								
OABSS	6.57 ± 0.68	5.31 ± 0.78	6.36 ± 0.43	4.70 ± 0.44	3.65 ± 0.38 ^##^	3.49 ± 0.42 ^##^	3.83 ± 0.42 ^##^	3.66 ± 0.38 ^##^
ICIQ-SF	11.47 ± 1.14	10.08 ± 1.07	11.26 ± 0.59	7.39 ± 0.56 ^+,##^	5.96 ± 0.60 ^##^	4.06 ± 0.45 ^##^	4.73 ± 0.48 ^##^	4.75 ± 0.59 ^##^
UDI-6	7.63 ± 0.62	6.77 ± 1.12	7.67 ± 0.48	4.63 ± 0.44 ^+,##^	3.53 ± 0.45 ^##^	3.05 ± 0.37 ^##^	3.06 ± 0.36 ^##^	2.91 ± 0.39 ^##^
IIQ-7	8.38 ± 1.30	7.00 ± 1.49	8.11 ± 0.67	4.36 ± 0.52 ^+,##^	3.07 ± 0.53 ^##^	2.19 ± 0.40 ^##^	2.67 ± 0.45 ^##^	2.52 ± 0.41 ^##^

Note: W, week; W4, once per week, 4-week of LiESWT; W8, once per week, 8-week of LiESWT; F1, 1-month follow up; F3, 3-month follow up; F6, 6-month follow up; OABSS, Overactive Bladder Symptom Scores; ICIQ-SF, International Consultation on Incontinence Questionnaire short form; UDI-6, Urinary Distress Inventory, Short Form; IIQ-7, Incontinence Impact Questionnaire, Short Form. Values are means ± SE. ^#^ *p* < 0.05; ^##^ *p* < 0.01 vs. W0. ^+^ *p* < 0.05; ^++^ *p* < 0.01 vs. sham group.

## Data Availability

The data that support the findings of this study are available from the corresponding author upon reasonable request.

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
