# Peer review of "Low Intensity Extracorporeal Shock Wave Therapy as a Novel Treatment for Stress Urinary Incontinence: A Randomized-Controlled Clinical Study"

_medicina, 2021, doi:10.3390/medicina57090947_

Round 1
Reviewer 1 Report
- Line 67 - “widely criticized” should be deleted because this is not true. You can write for example that it has significant rate of complications.
- Line 68-69 shold be deleted “Moreover, sling operation and bulking agent injection only enhance the periurethral support and have some anesthetic risks.”
- Line 69 “Therefore, novel therapies that can restore normal urethra function are urgently needed.” - I do not know about studies on treatment that can restore normal urethral function in women with SUI.
- Line 71-78: there should be a little more information about studies for example study on rat model, study did not analyze influence on urethra etc.
- Line 81: “multiple” should be deleted
- Table 1, inclusion criteria 5. “Urinary leakage >=2” - what does it mean?
- Table 2. Part: Serum parameters can be deleted - there is information in text that they were normal.
- Figure 4 can be deleted. This information may be useful in the leaflet for the patients.
- Line 266-270, ref. 5, 20, 21 - tape complications: please include statement from 20. Sounders: The rise in lawsuits does not reflect the acceptably low complication rates for slings for SUI reported in the literature. Please change 5 and 21 to newer articles about SUI tape complications.
- Macro results in Table 1 and 2 - can be round off to 1 place after point
- Physical parameters should be standardized (pulses/ second or Hz)
Very good article, but introduction and discussion should be more objective to surgery treatment of SUI and less promoting new method - it is to early to do this.
Author Response
Dear Dr. Editor and Dr. Reviewer 1,
We wish to thank the Editorial Board for the review of our manuscript entitled " Low intensity extracorporeal shock wave therapy as a novel treatment for stress urinary incontinence: a randomized-controlled clinical study ", which is being considered by the Medicina for publication.
By addressing every comment made by the reviewers, we have revised our manuscript. All the changes made in the manuscript are marked in red font.
We would like to thank you and the Editorial Board for the consideration and the intelligent review of our manuscript, which results in the revised manuscript of better quality.
Sincerely yours,
Yung-Shun Juan MD, PhD.
August 24th, 2021
Review 1:
Comments and Suggestions for Authors
Very good article, but introduction and discussion should be more objective to surgery treatment of SUI and less promoting new method - it is too early to do this.
- Line 67 - “widely criticized” should be deleted because this is not true. You can write for example that it has significant rate of complications.
Response: Thanks for your suggestion, we have been changed "Mid-urethral sling (MUS) surgery…of the high cure rate but with widely criticized for its early and late complications" to "Mid-urethral sling (MUS) surgery…of the high cure rate but has significant rate of complications" (please refer to page 2, Introduction section, lines 68-70).
- Line 68-69 should be deleted “Moreover, sling operation and bulking agent injection only enhance the periurethral support and have some anesthetic risks.”
Response: As suggested by the reviewer, we have deleted “Moreover, sling operation and bulking agent injection only enhance the periurethral support and have some anesthetic risks” in the Introduction section , lines 70-71, page 2.
- Line 69 “Therefore, novel therapies that can restore normal urethra function are urgently needed.” - I do not know about studies on treatment that can restore normal urethral function in women with SUI.
Response: According to two studies using a SUI rat model induced by vaginal balloon dilation (VBD), LiESWT can restore normal urethral function in women with SUI in the Introduction section (References #7 and #14). Indeed, “restore normal urethra function” is not a precise statement. We have changed “normal urethra function” to “normal pelvic floor and internal sphincter function” (please refer to page 2, Introduction section, line 72). We also added “Excessive hypermobility of the urethra and/or bladder neck resulted from pelvic floor degeneration and internal sphincter deficiency are the two main mechanisms of SUI.” on page 2, Introduction section, lines 62-64. Thanks for your professional recommendation.
- Line 71-78: there should be a little more information about studies for example study on rat model, study did not analyze influence on urethra etc.
Response: Thanks for your professional recommendation. Currently, there are two rat model studies using vaginal balloon dilation (VBD) to induce SUI. (Reference #7 and #14) We have added the descriptions with animal studies about the influence of LiESWT on SUI in the Introduction section (please refer to page 2, line 86-94) as ”A recent study using vaginal balloon dilation (VBD)-induced SUI rat model showed that rats treated with LiESWT (0.06 mJ/mm2, and 300 shocks at 3 pulses/second) following VBD had significantly higher leak point pressure (LPP) relative to those receiving VBD only. LiESWT improved LPP and ameliorated SUI by promoting urethral sphincter regeneration, angiogenesis, and progenitor cell recruitment [7]. With the same energy and course of LiESWT, another VBD-induced SUI rat study revealed LiESWT treatment increased amount of urethral and vaginal smooth and striated muscle content and increased thickness of the vaginal wall after LiESWT treatment [14]. Higher LPP was also found in the LiESWT group [14].”
- Line 81: “multiple” should be deleted
Response: As suggested by the reviewer, we have deleted the word “multiple” in the Introduction section on line 84, page 2.
- Table 1, inclusion criteria 5. “Urinary leakage >=2” - what does it mean?
Response: The inclusion criteria 5. “Urinary leakage >=2” means “the urinary leakage ≥ 2 g”. We have been revised the “Urinary leakage ≥ 2” to “Urinary leakage ≥ 2g” in Table 1, inclusion criteria #5 on line 118, page 3.
- Table 2. Part: Serum parameters can be deleted - there is information in text that they were normal.
Response: The serum parameters in Table 2 help readers understand the detailed information of patients’ characteristics, we would keep those information in Table 2.
- Figure 4 can be deleted. This information may be useful in the leaflet for the patients.
Response: Thanks for your suggestion. Since the figure 4 was a short graphic abstract of current study, it would help readers understand the proposed potential therapeutic effect of LiESWT on SUI, we intend to keep Figure 4 in the manuscript.
- Line 266-270, ref. 5, 20, 21 - tape complications: please include statement from 20. Sounders: The rise in lawsuits does not reflect the acceptably low complication rates for slings for SUI reported in the literature. Please change 5 and 21 to newer articles about SUI tape complications.
Response: Thanks for your suggestion, we have replaced ref.5 and ref.21 with new reference (Lin, Y.H. et al., 2021) (please refer to page 2, Introduction section, line 70 and page 9, Discussion section, line 294). We also have added more information on the complication rates for slings surgery for SUI, as “The rise in lawsuits does not reflect the acceptably low complication rates for slings for SUI reported in the literatures. Reported complications included bladder injury (0.8-11.4%), vaginal injury (0.8-15%), hemorrhage (0.7-5.5%), urinary tract infections (0.9-33%), urine retention (0.8-11.4%), and mesh erosion (0-10%)” (please refer to page 9, Discussion section, lines 291-294).
Reference list:
Ref. 5: Nitti, V.W. Complications of midurethral slings and their management. Can Urol Assoc J. 2012, 6, S120-122
Ref. 21: Daneshgari, F.; Kong, W.; Swartz, M. Complications of mid urethral slings: important outcomes for future clinical trials. J Urol. 2008, 180, 1890-1897
New reference: Lin, Y.H.; Lee, C.K.; Chang, S.D.; Chien, P.C.; Hsu, Y.Y.; Tseng, L.H. Focusing on long-term complications of mid-urethral slings among women with stress urinary incontinence as a patient safety improvement measure A protocol for systematic review and meta-analysis. Medicine. 2021, 100
- Macro results in Table 1 and 2 - can be round off to 1 place after point
Response: As suggested by the reviewer, the macro results in Table 1 and Table 2 have been round off to 1 place after point.
- Physical parameters should be standardized (pulses/ second or Hz)
Response: As suggested by the reviewer, we have standardized the word “Hz” to “pulses/ second” in the Introduction section on line 87, page 2 and Discussion section on line 269, page 9.

Reviewer 2 Report
The authors performed a RCT to evaluate the effects of LiESWT on stress urinary incontinence
Please, elaborate on the details of the sham group. How was the absence of transmission or energy ensured?
With such small sample size, the study only proves that LiESWT is a safe procedure. However, the efficacy of this therapy for urinary incontinency requires a larger cohort
Table 2 with sociodemographic information is not testing a hypothesis. Therefore, p values should not be displayed
Out of the parameters evaluated in table 3, only average voided volume and urgency consistently showed a significant improvement compared to W0. However, how clinically relevant is an improvement of 10-13 ml in average voided volume or a 1 unit reduction in nocturia events. Moreover, the trial was targeting stress urinary incontinence. Nocturia has nothing to do with SUI. In the same context, although significant, the improvement in the questionnaires are clinically meaningless.
Statistically the authors have to prove that the evaluated parameters are normally distributed in the evaluated population. If paired t test or student t test are going to be performed. It would be unexpected to find that these parameters are normally distributed in the population. If parameters show not to be normally distributed, non-parametrical statistical test must be performed. Also, the number of t-test performed displays a lack of a clear hypothesis and research question. Given the number of t test shown in this study it seems that the authors are testing multiple hypotheses at the same time, which in my opinion is wrong. In the same context, if multiple hypotheses are going to be tested through multiple t test or other non-parametrical statistical test, the p values must be adjusted for multiple comparisons given the higher chance of alpha error.
As I mentioned before this trial would only work for showing safety of this novel technique. The samples size prevents to draw any conclusion in the context of efficacy which is what the conclusion section seems to focus on.
There are several methodological flaws in this study.
Author Response
Dear Dr. Editor and Dr. Reviewer 2,
We wish to thank the Editorial Board for the review of our manuscript entitled " Low intensity extracorporeal shock wave therapy as a novel treatment for stress urinary incontinence: a randomized-controlled clinical study ", which is being considered by the Medicina for publication.
By addressing every comment made by the reviewers, we have revised our manuscript. All the changes made in the manuscript are marked in red font.
We would like to thank you and the Editorial Board for the consideration and the intelligent review of our manuscript, which results in the revised manuscript of better quality.
Sincerely yours,
Yung-Shun Juan MD, PhD.
August 24th, 2021
Review 2:
Comments and Suggestions for Authors
The authors performed a RCT to evaluate the effects of LiESWT on stress urinary incontinence. There are several methodological flaws in this study.
- Please, elaborate on the details of the sham group. How was the absence of transmission or energy ensured?
Response: The sham group use air pad to block energy transmission during treatment, but the machine still emitted a shock wave generation. However, most participants still felt the pulsive vibration of the probe. We have added the related descriptions as “The sham group used an air pad probe to block energy transmission to the tissue, but the machine still emitted shock wave generation.” in the Materials and Methods section (please refer to page 4, lines 136-137).
The air pad used was shown in the below figure.
- With such small sample size, the study only proves that LiESWT is a safe procedure. However, the efficacy of this therapy for urinary incontinency requires a larger cohort. As I mentioned before this trial would only work for showing safety of this novel technique. The samples size prevents to draw any conclusion in the context of efficacy which is what the conclusion section seems to focus on.
Response: As mentioned by the reviewer, the sample size of current study was small. We have mentioned this point in the Discussion section on lines 329 to 331, page 10 “Our study had several limitations. First, patients were limited in small numbers. People benefiting from LiESWT treatment have yet to be confirmed in larger prospective studies.”
- Table 2 with sociodemographic information is not testing a hypothesis. Therefore, p values should not be displayed.
Response: As suggested by the reviewer, the p values in Table 2 have been deleted on line 178, page 5.
Table 2. Baseline characteristics of stress urinary incontinence (SUI) population
|
Parameter |
Sham (Mean ± SE) |
LiESWT (Mean ± SE) |
Range |
|
Physical parameter |
  |
  |
  |
|
Female age (years) |
52.8 ± 1.9 |
53.8 ± 1.5 |
20 - 75 |
|
Height (cm) |
158.0 ± 1.6 |
158.9 ± 0.9 |
|
|
Weight (kg) |
59.4 ± 1.9 |
61.2 ± 1.3 |
|
|
BMI (kg/m2) |
23.8 ± 0.7 |
24.2 ± 0.4 |
18.5 - 24 |
|
Waistline (cm) |
87.2 ± 2.8 |
85.6 ± 1.4 |
|
|
Systolic pressure (mmHg) |
124.4 ± 3.0 |
121.4 ± 2.9 |
100 - 120 |
|
Diastolic pressure (mmHg) |
77.9 ± 2.3 |
74.5 ± 1.7 |
60 - 80 |
|
MAP |
93.4 ± 2.3 |
90.1 ± 2.0 |
70 - 110 |
|
Serum parameter |
|
|
|
|
HbA1C (%) |
5.67 ± 0.12 |
5.63 ± 0.06 |
4 - 6 |
|
AC sugar (mg/dl) |
101.7 ± 3.7 |
101.0 ± 1.6 |
65 - 109 |
|
BUN (mg/dl) |
12.54 ± 0.80 |
12.35 ± 0.45 |
8 - 20 |
|
Creatinine (mg/dl) |
0.67 ± 0.03 |
0.68 ± 0.02 |
0.44 - 1.03 |
|
GOT(AST) (IU/L) |
22.5 ± 1.0 |
23.5 ± 1.3 |
10 - 42 |
|
GPT(ALT) (IU/L) |
23.8 ± 2.2 |
23.9 ± 2.2 |
10 - 40 |
|
Triglycerides (mg/dl) |
92.1 ± 12.3 |
96.9 ± 9.0 |
35 - 160 |
|
Cholesterol (mg/dl) |
214.2 ± 8.2 |
203.7 ± 6.7 |
140 - 200 |
|
HDL (mg/dl) |
61.5 ± 3.6 |
57.2 ± 2.2 |
29 - 85 |
|
LDL (mg/dl) |
134.9 ± 8.3 |
124.9 ± 6.3 |
0 - 130 |
- Statistically the authors have to prove that the evaluated parameters are normally distributed in the evaluated population. If paired t test or student t test are going to be performed. It would be unexpected to find that these parameters are normally distributed in the population. If parameters show not to be normally distributed, non-parametrical statistical test must be performed. Also, the number of t-test performed displays a lack of a clear hypothesis and research question. Given the number of t test shown in this study it seems that the authors are testing multiple hypotheses at the same time, which in my opinion is wrong. In the same context, if multiple hypotheses are going to be tested through multiple t test or other non-parametrical statistical test, the p values must be adjusted for multiple comparisons given the higher chance of alpha error.
Response: As suggested by the review, we have compared Student's t test and paired t-test with one-way ANOVA following post-hoc Tukey’s test as below. We also modified our Figure 2, Figure 3, and Table 3 accordingly. The following information has been added to the Materials and Methods section on page 5, lines 160-165, as “Paired t-test and one-way analysis of variance were used to perform a repeated measurement analysis for intragroup before/after treatment [19]. The paired t-test was performed in the sham group (W0 vs. W4). The post-hoc Tukey’s honestly significant difference tests were used to make comparison between the LiESWT subgroups and to calculate p-values for comparison [19]. The student t-test was performed for the intergroup comparison (the sham group vs. the LiESWT group).” We also modified our Abstract section (lines 44 and 47 on page 1), Result section (lines 205-209, 218 and 221 on page 7) and Discussion section (lines 252 and 255 on page 8).
Reference:
- Tervonen, T.; Karjalainen, K. Periodontal disease related to diabetic status - A pilot study of the response to periodontal therapy in type 1 diabetes. Journal of Clinical Periodontology. 1997, 24, 505-510.
Fig 2a and 2b
- Student’s t-test, Paired t-test and One-way ANOVA
Fig 3a and 3b
- Student’s t-test, Paired t-test and One-way ANOVA
Table 3.
- Student’s t-test, Paired t-test and One-way ANOVA
- Out of the parameters evaluated in table 3, only average voided volume and urgency consistently showed a significant improvement compared to W0. However, how clinically relevant is an improvement of 10-13 ml in average voided volume or a 1 unit reduction in nocturia events. Moreover, the trial was targeting stress urinary incontinence. Nocturia has nothing to do with SUI. In the same context, although significant, the improvement in the questionnaires are clinically meaningless.
Response: After re-analyzing the comparison between LiESWT subgroups and calculating the p-values for comparison by one-way ANOVA following post-hoc Tukey’s test, LiESWT significant improvement about 37 ml in average voided volume and a 1.03 unit reduction in urgency, but there was no significant difference in nocturia events (please refer to Table 3).
Indeed, symptoms of SUI and OAB include nocturia are on different arms of urinary incontinence. However, we also found the interesting result that LiESWT for SUI can also ameliorate OAB symptoms. The main reason why LiESWT for SUI can also ameliorate OAB symptoms is LiESWT can restore pelvic floor function and thereafter improve conscious control of the bladder by activating the part of the brain responsible for the voluntary urinary inhibition reflex. In our previous Er:YAG vaginal laser treatment for SUI, OABSS also significantly decreased following vaginal laser treatment. We have discussed about this in our 4th paragraph of discussion section (please refer to page 9, lines 301-317) and added another statement, as” Interestingly, LiESWT also ameliorated overactive bladder (OAB) symptoms during follow up.” in the introduction section (please refer to page 2, lines 96-97). We also add another statement in the Discussion section “The main reason why LiESWT for SUI can also ameliorate OAB symptoms is LiESWT can restore pelvic floor function and thereafter improve conscious control of the bladder by activating part of the brain responsible for the voluntary urinary inhibition reflex.” (please refer to page 10, lines 305-308) and “In our previous Er:YAG vaginal laser treatment for SUI study, OABSS also significantly decreased following vaginal laser treatment.” (please refer to page 10, lines 313-314).

Round 2
Reviewer 2 Report
My comments about further explanation of the sham group have been addressed by the authors
About my comment “With such small sample size, the study only proves that LiESWT is a safe procedure. However, the efficacy of this therapy for urinary incontinency requires a larger cohort. As I mentioned before this trial would only work for showing safety of this novel technique. The samples size prevents to draw any conclusion in the context of efficacy which is what the conclusion section seems to focus on” the authors recognize that the study was performed in a small sample size. In this sense, I consider the title of the study misleading as it hints that is indeed a study that describe a novel therapy for SUI. the title should go different and should emphasize that this method is safe, period. This is not yet a therapy for SUI. Indeed, it is far from being effective and it’s experimental
My comments about table 2 have been addressed by the authors
About my comments “Statistically the authors have to prove that the evaluated parameters are normally distributed in the evaluated population. If paired t test or student t test are going to be performed. It would be unexpected to find that these parameters are normally distributed in the population. If parameters show not to be normally distributed, non-parametrical statistical test must be performed. Also, the number of t-test performed displays a lack of a clear hypothesis and research question. Given the number of t test shown in this study it seems that the authors are testing multiple hypotheses at the same time, which in my opinion is wrong. In the same context, if multiple hypotheses are going to be tested through multiple t test or other non-parametrical statistical test, the p values must be adjusted for multiple comparisons given the higher chance of alpha error” — According to Assel et al “The more questions you ask, the more likely you are to get a spurious answer to at least one of them. For example, if you report p values for five independent true null hypotheses, the probability that you will falsely reject at least one is not 5%, but >20%. Although formal adjustment of p values is appropriate in some specific cases, such as genomic studies, a more common approach is to simply interpret p values in the context of multiple testing. For instance, if an investigator examines the association of 10 variables with three different endpoints, thereby testing 30 separate hypotheses, a p value of 0.04 should not be interpreted in the same way as if the study tested only a single hypothesis with a p value of 0.04”1. Also, “A common analysis is to conduct a paired t test comparing, say, erectile function in older men at baseline with erectile function after 5 yr of follow-up. The null hypothesis here is that “erectile function does not change over time,” which is known to be false. Investigators are encouraged to focus on estimation rather than on inference, reporting, for example, the mean change over time along with a 95% CI”1. Therefore, multiple p values without adjusting for multiple comparison is huge methodological flaw and as mentioned before the risk of alpha error increase dramatically. Also, multiple t test to evaluate results overtime it is strongly recommended to compare the mean change over time.
- Assel M, Sjoberg D, Elders A, Wang X, Huo D, Botchway A, et al. Guidelines for Reporting of Statistics for Clinical Research in Urology. European urology. 2019;75(3):358-67.
About my comment “Out of the parameters evaluated in table 3, only average voided volume and urgency consistently showed a significant improvement compared to W0. However, how clinically relevant is an improvement of 10-13 ml in average voided volume or a 1 unit reduction in nocturia events. Moreover, the trial was targeting stress urinary incontinence. Nocturia has nothing to do with SUI. In the same context, although significant, the improvement in the questionnaires are clinically meaningless” — Again, the title is misleading, are we testing OAB or SUI, if both, why does the title only mention SUI? Also, nocturia as per the international continence society is defined as “Stress urinary incontinence (SUI) is defined by the International Continence Society (ICS) as “the complaint of any involuntary loss of urine on effort or physical exertion (e.g sporting activities) or on sneezing or coughing” [1]. This is the type of incontinence most commonly reported by patients with chronic chest conditions.” no mention of nocturia. Is correct that nocturia is part of OAB however, my comment was clearly about SUI. I still think that 1 unit reduction in nocturia is clinically meaningless.